

# Network subgraph-based approach for analyzing and comparing molecular networks

Chien-Hung Huang[1,*], Efendi Zaenudin[2,3], Jeffrey J.P. Tsai[3], Nilubon Kurubanjerdjit[4] and Ka-Lok Ng[3,5,6,*]

[1] Department of Computer Science and Information Engineering, National Formosa University, Yun-Lin, Taiwan
[2] National Research and Innovation Agency, Bandung, Jawa Barat, Republic of Indonesia
[3] Department of Bioinformatics and Medical Engineering, Asia University, Taichung, Taiwan
[4] School of Information Technology, Mae Fah Luang University, Chiang Rai, Thailand
[5] Center for Artificial Intelligence and Precision Medicine Research, Asia University, Taichung, Taiwan
[6] Department of Medical Research, China Medical University Hospital, China Medical University, Taichung, Taiwan
[*] These authors contributed equally to this work.

Corresponding author
Ka-Lok Ng, ppiddi@gmail.com

## ABSTRACT

Molecular networks are built up from genetic elements that exhibit feedback interactions. Here, we studied the problem of measuring the similarity of directed networks by proposing a novel alignment-free approach: the network subgraph-based approach. Our approach does not make use of randomized networks to determine modular patterns embedded in a network, and this method differs from the network motif and graphlet methods. Network similarity was quantified by gauging the difference between the subgraph frequency distributions of two networks using Jensen–Shannon entropy. We applied the subgraph approach to study three types of molecular networks, *i.e.*, cancer networks, signal transduction networks, and cellular process networks, which exhibit diverse molecular functions. We compared the performance of our subgraph detection algorithm with other algorithms, and the results were consistent, but other algorithms could not address the issue of subgraphs/motifs embedded within a subgraph/motif. To evaluate the effectiveness of the subgraph-based method, we applied the method along with the Jensen–Shannon entropy to classify six network models, and it achieves a 100% accuracy of classification. The proposed information-theoretic approach allows us to determine the structural similarity of two networks regardless of node identity and network size. We demonstrated the effectiveness of the subgraph approach to cluster molecular networks that exhibit similar regulatory interaction topologies. As an illustration, our method can identify (i) common subgraph-mediated signal transduction and/or cellular processes in AML and pancreatic cancer, and (ii) scaffold proteins in gastric cancer and hepatocellular carcinoma; thus, the results suggested that there are common regulation modules for cancer formation. We also found that the underlying substructures of the molecular networks are dominated by irreducible subgraphs; this feature is valid for the three classes of molecular networks we studied. The subgraph-based approach provides a systematic scenario for analyzing, compare and classifying molecular networks with diverse functionalities.

## INTRODUCTION

Network comparison is an important and well-studied subject in bioinformatics. Many molecular biology networks are directed networks (digraphs), such as gene regulatory networks, signal transduction networks, cellular processes, metabolic networks etc. In the area of computational molecular biology, it is useful to compare networks with each other, because if the properties of a given network are known, one can transfer this information to the other network; such as, structural study of metabolic networks for single cell organism (*Zhu & Qin, 2005*), inferring phylogenetic tree for metabolic networks (*Heymans & Singh, 2003*), grouping different types networks; such as, cell signaling, metabolic and transcriptional regulatory networks (*Aparìcio, Ribeiro & Silva, 2015*).

### Alignment-based methods and alignment-free methods

Alignment-based methods are mainly used to compare whether the nodes and edges of two networks are similar, and to identify conserved modules. Alignment-free methods do not need to consider the node identity and network size, so they may not find conserved regions (*Yaveroğlu, Milenković & Pržulj, 2015*); instead, the methods can extract conserved topological similar regions.

There are a few methods that can be used to analyze digraphs (*Tantardini et al., 2019*); including (i) global statistics (*Pržulj, 2007*), (ii) portrait divergence (PD) (*Bagrow & Bollt, 2019*), (iii) graphlet (*Sarajlić et al., 2016*) and (iv) algorithmic complexity of network motifs (*Zenil, Kiani & Tegnér, 2013*). Methods (i) to (iii) use graph theory metrics, whereas (iv) uses information contents of the motifs to compare networks. Each method has its advantages and limitations.

Networks with similar global statistics do not necessary mean similar network architecture (*Pržulj, 2007*). The PD method defines three probability distributions to characterize a graph: (i) the probability that two points in the network are connected, (ii) the probability that the distance between the two nodes is $L$, and (iii) the probability that one of the nodes is connected to a $k$-$1$ nodes at a distance of $L$. Then, the graph invariant (the network portrait) is defined by taking the normalized product of these three probability distributions. Level of network similarity is given by the Jensen–Shannon entropy ($H_{JS}$) of the two network portraits. The authors demonstrated that their method was able to distinguish the protein interaction, neuroscience, and social science networks, but only a few networks are considered.

The use of graphlets was introduced by Przulj to perform network comparisons (*Przulj, 2007*; *Yaveroglu et al., 2014*). Graphlets are connected small graphs. Each node in the graphlet can be divided into different categories depending on its connection to other nodes in the network. Nodes in the same category belong to the same orbit. To study directed networks, the concept of the graphlet was extended to directed graphlets by considering the in- and out-degree. Directed graphlets were demonstrated to be superior
for comparing directed networks (*Sarajlić et al., 2016*) and effective in studying brain networks (*Trpevski et al., 2016*). The graphlet method compares two directed networks by decompose the network into three-node and four-node graphlets, and calculate the Euclidean distance between their Directed Graphlet Correlation Matrix (*Sarajlić et al., 2016*). *Sarajlić et al. (2016)* applied the directed graphlet degree concept to predict the biological function of enzymes according to the similarity of their connection patterns in a metabolic network. *Trpevski et al. (2016)* analyzed the brain network by using the (i) signature vector of the vertex (brain region) and (ii) graphlet correlation matrix of the network to infer the excitatory/inhibitory and causal patterns, respectively, in the effective brain networks.

New methods have been developed to perform directed network comparisons, including expanding the definition of graphlets (*Martin et al., 2016*) and using graphlet-based metrics (GBM) (*Martin et al., 2017*). *Martin et al. (2016)* introduced the rate of graphlet reconstruction and REC graphlet degree (RGD) to compare gene regulatory networks (from *Escherichia coli*) under a specific condition. In another study, *Martin et al. (2017)* applied GBM to assess the topological similarity between networks (from *E. coli*) under different biological conditions. Previous research on network analyses focused on graphlets composed of three nodes only; however, the extension of graphlets with four nodes is still limited (*Trpevski et al., 2016*; *Martin et al., 2016*; *Martin et al., 2017*).

Furthermore, *Zenil, Kiani & Tegnér (2013)* applied the Block Decomposition Method to estimate algorithmic probability of the four-node motifs of a network, the authors demonstrated that their method correctly distinguish the developmental genetic network and the signal transduction network, the performance is better than the compression algorithm, BZIP2.

Many tools (*Tran et al., 2015*; *Meira et al., 2018*; *Meira et al., 2014*; *Wernicke & Rasche, 2006*; *Omidi, Schreiber & Masoudi-Nejad, 2009*; *Meira et al., 2014*) have been developed to detect 'statistically significant' network motifs. The *acc-Motif* algorithm can identify network motifs with a size of up to five nodes. Later, the algorithm was improved to find motifs for up to six nodes (*Meira et al., 2018*). We noted that the tool *LoTo* identifies motifs for up to three nodes but not for four nodes. But these tools may not be able to detect the complete set of motifs, because the predicted patterns are not statistically significant. This suggests that motif-finding tools have limitations in our earlier work (*Huang et al., 2020*), as they cannot enumerate all the network motifs embedded in a network due to the use of an arbitrary threshold; *i.e.*, *p*-value is larger than 0.05. Also, it is known that the time complexity of identifying $N$-node motifs in a large network is an NP-complete problem (*Kim et al., 2013*).

## The network subgraph-based approach and network comparison

*Mowshowitz (1968)*, who developed a method to address the problem of gauging the relative complexity of graphs. Drawing on that, in our previous work (*Huang et al., 2020*) we propose the network subgraph-based approach, treat the network subgraphs' pattern *exactly* the same as the network motifs. but not make use of the randomization definition to extract the subgraphs embedded in a network.

Network similarity can be quantified by using the information-theoretic quantity Kullback–Leibler entropy ($H_{KL}$). In 2011, *Kugler et al. (2011)* introduced the use of graph prototyping for network comparison. They showed that in three out of the five graph distance measures used, the group of prostate cancer networks differed significantly from the group of benign networks. $H_{KL}$ is an asymmetric quantity, one can define a symmetric quantity, $H_{JS}$, to gauge network similarity.

Our method is an alignment-free approach. It is different from some earlier studies (*Sarajlić et al., 2016*; *Trpevski et al., 2016*; *Martin et al., 2016*; *Martin et al., 2017*) in that we employed a subgraph instead of a graphlet to dissect network topologies.

In the 'Methods' section, we present our approach, introduce network comparison method and describe setting up simulation experiments to test our method. In the 'Results' section, we present the effectiveness of our approach on simulated data, and its use to evaluate similarities amongst three categories of molecular biology networks. We also address the biological meanings of topological similar molecular networks. In the 'Conclusion' section, we elaborate the key findings in this work, and suggest biological applications of our method in future study.

## METHODS

### Network subgraph-based approach

*Mowshowitz (1968)* proposed that a finite graph ($N$ nodes and $E$ edges) can be decomposed into equivalence classes ($C$ classes); each class contains $n_i$ nodes, and a probability is assigned to each class, *i.e.*, $p_j = n_j/N$. There are many ways of partitioning the set of nodes of a graph, one way to obtain a decomposition is to identify the orbits of a graph. The orbits of a graph can be identified by calculating the degree of the nodes, point-deleted neighborhood degree vector and betweenness centrality (*Mowshowitz & Mitsou, 2009*). The subgraph-based approach we developed does not rely on determining the orbits of a graph; hence, it is a different one.

### Adjacency matrix

Given a network subgraph, one can construct an adjacency matrix $A$, with matrix elements "0" and "1" to represent the absence and presence of connections among the nodes, respectively. Each subgraph can be represented by a decimal. This can be achieved by arranging all the entries in the adjacency matrix by row major order into one binary string and then convert it to decimal, each subgraph can be denoted by an unique decimal value, called graph ID.

### Subgraph identification, network comparison and simulation experiments

Previously, we developed a subgraph detection algorithm, *PatternFinder* (*Huang et al., 2020*), to identify three-node subgraphs and four-node subgraphs embedded in cancer networks, STN, and cellular processes. A brief description of the algorithm, *PatternFinder*, was provided in the Appendix section. Also, we point out that *PatternFinder* is an exhaustive search algorithm, which allowed us to detect the complete set of subgraphs and subgraphs within a subgraph, it is not intend for large-scale network analysis.
In information theory, a number of quantities can be used to characterize the distance between two probability distributions. For instance, one can use the Kullback–Leibler entropy, $H_{KL}$, also known as cross-entropy (*Capra & Singh, 2007*) to quantify the distance. Given two discrete probability distributions, $X$ and $Y$, the Kullback–Leibler entropy of $X$ with respect to $Y$ is defined by

$$H_{KL}(X||Y) = -\sum_i X_i \log \frac{X_i}{Y_i}. \tag{1}$$

$H_{KL}$ is asymmetric under the interchange of $X$ and $Y$. In 2011, *Kugler et al. (2011)* introduced the use of $H_{KL}$ for network comparison. They showed that in three out of the five graph distance measures used, the group of prostate cancer networks differed significantly from the group of benign networks.

One can symmetrize $H_{KL}$ by adding the term $H_{KL}(Y||X)$. We applied a similar quantity, $H_{JS}$, to gauge network similarity. $H_{JS}$ is a symmetric function and is used to measure the distance between the two subgraph probability distributions $X$ and $Y$ for networks $N_X$ and $N_Y$, respectively. $H_{JS}$ is defined as follows:

$$\boldsymbol{H_{JS}} = \frac{1}{2}[\boldsymbol{H_{KL}(X||Z) + H_{KL}(Y||Z)}]. \tag{2}$$

where $Z = \frac{1}{2}(X+Y)$.

Let $X^{(n)}$ and $Y^{(n)}$ be the $n$-node subgraph probability distributions obtained by using *PatternFinder* for networks $N_X$ and $N_Y$, respectively, where $n = 3$ or $4$. If $n$ equals to 3, $X^{(3)}$ and $Y^{(3)}$ are compose of thirteen components; obviously, $X^{(4)}$ and $Y^{(4)}$ are compose of 199 components. Then, the three-node subgraph Jensen–Shannon entropy measure for networks $N_X$ and $N_Y$, $H_{JS}^{(3)}$ is given by

$$H_{JS}^{(3)} = \frac{1}{2}\left[\boldsymbol{H_{KL}}\left(\boldsymbol{X}^{(3)}||\boldsymbol{Z}^{(3)}\right) + \boldsymbol{H_{KL}}\left(\boldsymbol{Y}^{(3)}||\boldsymbol{Z}^{(3)}\right)\right] \tag{3}$$

where $Z^{(n)} = \frac{1}{2}(X^{(n)} + Y^{(n)})$, with $n = 3$. A similar expression for the four-node subgraph Jensen–Shannon entropy measure, $H_{JS}^{(4)}$, can be obtained simply by substituting $X^{(4)}$, $Y^{(4)}$ and $Z^{(4)}$ into Eq. (3).

The unique feature of $H_{JS}$ is that it measures the similarity between two networks in terms of the underlying architecture of the networks rather than the identities of the nodes. In other words, network similarity is measured without referring to the genetic identities of a subgraph. The square root of $H_{JS}$ is a metric called the Jensen–Shannon distance (*Endres & Schindelin, 2003*).

$H_{JS}$ has been used in many applications, such as (i) predicting functionally important amino acids from sequence conservation, (ii) pattern recognition in bioinformatics (*Loog et al., 2011*), (iii) predicting important non-conserved residues in protein sequences (*Gültas et al., 2014*), (iv) analyzing DNA sequences (*Grosse et al., 2002*), and (v) measuring the distance between random graphs (*Wong & You, 1985*). *Wong & You (1985)* proposed a distance metric between two random graphs based on the smallest change in Shannon entropy before and after merging the two random graphs.

In order to verify the effectiveness of the subgraph-based approach along with the $H_{JS}$ metric, we used the 'igraph' package (*Csardi & Nepusz, 2006*) (https://igraph.org/) to
**Table 1  Description of the six network models, 'igraph' parameter settings and meaning of the parameters.**

| Network type | Description of the network (igraph) | parameters used in 'igraph' |
| --- | --- | --- |
| Random graph | This model is very simple, every possible edge is created with the same constant probability | Prob = 0.0188, directed = TRUE |
| Scale-free network | The BA-model is a very simple stochastic algorithm for building a graph | Power = 2, $m = 3$, zero.appeal = 1, directed=TRUE, algorithm=psumtree |
| Small world | Generate a graph according to the Watts–Strogatz network model | Dim = 1, Nei = 25, $p = 0.6$, directed = TRUE |
| Geometric random graph | Generate a random graph based on the distance of random point on a unit square | radius = 0.15, torus = TRUE, coords = TRUE, directed = TRUE |
| Aging random graph | This function creates a random graph by simulating its evolution. Each time a new vertex is added it creates a number of links to old vertices and the probability that an old vertex is cited depends on its in-degree (preferential attachment) and age | pa.exp = 1, aging.exp = −1, aging.bin = 1,000, directed = TRUE |
| Citation random graph | creates a graph, where vertices age, and gain new connections based on how long ago their last citation happened | edges = 1, age_bins=nodes/100 agebins,pref = (1:(agebins + 1)) 3, directed = TRUE |

generate six different network models (Random graph, Scale-free network, Small world, Geometric random graph, Aging random graph, Citation random graph, using three different number of nodes: 300, 400 and 500, and two types of edge density: 2% and 6%, each network repeated three times, *i.e.*, a total of 6*3*2*3 = 108 networks) and then used MST-kNN (*Arefin et al., 2014*) to perform clustering classification. MST-kNN is an unsupervised graph-based clustering classifier based on Jensen–Shannon divergence and graph partition algorithm, it was utilized to classify the authorship of drama and poems. Table 1 described these six models, 'igraph' parameter settings (*Csardi & Nepusz, 2006*), and meaning of the parameters.

## Input datasets—cancer networks, signal transduction networks (STN) and cellular processes

Network information was retrieved from the KEGG database (*Nakaya et al., 2013*). After manual inspection, we removed networks composed of separate components, such as "chemical carcinogenesis", "microRNAs in cancer", "two-component system", and "viral carcinogenesis". In addition, we collected the networks labelled with "signaling pathway", grouped them together, and called them "signal transduction networks (STN)". We note that STN range across different families of molecular networks recorded by KEGG, including "endocrine system", "immune system", and "signal transduction". We compiled three major types of molecular networks, *i.e.*, 17 cancer networks, 45 STN, and nine cellular processes. Names of these three types of molecular networks are listed in File S1.

**Table 2** Summary of the parameters of three simulation experiments, including six network models, network sizes, edge densities, total number of networks, and classification accuracy.

| simulation | Network type | Network sizes | Edge density | Total number of networks | Accuracy |
| --- | --- | --- | --- | --- | --- |
| I | Random graph, Scale-free network, Small world, Geometric random graph, Aging random graph, Citation random graph | 300, 400, 500 | 2% | 54 | 100% |
| II | Random graph, Scale-free network, Small world, Geometric random graph, Aging random graph, Citation random graph | 300, 400, 500 | 6% | 54 | 100% |
| III | Small world, Aging random graph, Citation random graph | 300, 400, 500 | 2% & 6% | 54 | 100% |

We downloaded the KGML files of the 71 networks from the KEGG database and used the Cytoscape plug-in tools KEGGScape (*Nishida et al., 2014*) and KEGGparser (*Arakelyan & Nersisyan, 2013*) to obtain node and edge information for those networks. Real-world molecular networks are composed of thousands of genes, which are larger than the networks we analyzed; however, the regulatory and feedback interaction information among thousands of genes are not available in KEGG, yet it can be analyzed once the data are available.

## RESULTS

### Comparison of tools—*PatternFinder*, *acc-Motif* and *LoTo*

To evaluate the performance of *PatternFinder* in enumerating all the three-node subgraphs, we have demonstrated the usefulness of tool in Table 2 of our previous study (*Huang et al., 2020*).

Essentailly speaking, *PatternFinder* is able to identify the complete set of subgraphs, whereas *acc-Motif* identifies relatively few network motifs. Furthermore, *acc-Motif* is unable to identify some of the four-node subgraphs for a given network due to the fact that those subgraphs are not statistically significant. Also, we noted that the tool *LoTo* identifies motifs for up to three nodes but not for four nodes. The above results suggest that motif-finding tools may have certain limitations as they cannot enumerate all possible substructures of a network. We provided Files S2 and S3 to help the reader relate (i) the subgraphs' decimals and their graphical representation, and (ii) the *acc-Motif* IDs and their graphical representation.

Given the three-node subgraphs and four-node subgraphs identified by *PatternFinder*, the normalized frequency distributions of the thirteen three-node subgraphs and 199 four-node subgraphs were determined; hence, network similarity was quantified by using the information-theoretic quantity $H_{JS}$.

### Classification of network models using subgraph-based approach

To examine the effectiveness of the subgraph-based approach, we applied the method along with $H_{JS}$ to classify six network models. We considered three types of simulations (Table 2). Simulations I and II consider six network models, each model with three different node numbers, but with the same edge density, *i.e.*, 2% and 6% respectively; and repeat the

simulation for three trials, therefore, each simulation composes of $6 \times 3 \times 3 = 54$ networks need to classify. The third simulation considered three network models; with different numbers of nodes and edge densities, again there are 54 networks ($3*3*2*3 = 54$) to classify.

Given the 54 networks, we performed the following: (i) applied *PatterFinder* to extract the three-node subgraph probability distribution for each network, (ii) computed the pairwise $H_{JS}$ distance matrix (a matrix with dimension $54 \times 54$, File S4 summarizes the $H_{JS}$ values for the three simulations), and (iii) used the MST-kNN package to do cluster analysis. The results of classification achieve 100% accuracy for the three simulations (Table 2 last column & Figs. 1A–1C). For example, Fig. 1A shown that the 54 networks are correctly classified into six network models. Orange color circles in Fig. 1A denote "Geometric Random Graph", the numbers listed inside the circles start from 1 to 9, which are the first nine networks in the pairwise $H_{JS}$ distance matrix. Similarly, the blue colored circles included numbers run from 10 to 18, which represent "Random Graph". We checked the numbers listed in the other four different colored circles, which correctly denote the other four types of network models. The result of Fig. 1C shows that given the $H_{JS}$ distance matrix, the MST-kNN classifier correctly classify the six possibilities (three network models along with two different edge densities). Compared with the results presented in *Sarajlić et al. (2016)*, our classification accuracy is better; but the network we analyzed is relatively small, the number of nodes is not more than 500.

File S5 provided the codes for (i) generating the six models, (ii) calculating the $H_{JS}$, and (iii) performing cluster analysis using MST-kNN.

To address the concern why our method can distinguish networks with similar global topology, we provide a detail discussion in File S6. In essence, we plot the cumulative probability functions ('cpfs') of the three-node subgraphs for the first trial; so there are 18 'cpfs'. As shown in the Fig. S6.1, the small-world and scale-free networks 'cpfs' can be clearly distinguished from the rest. Among the other four random networks, the 'cpfs' of aging random network and citation random network are quite close but with minor difference, as shown in Fig. S6.2; thus, indicates the effectiveness of our method.

## Cancer networks

We performed a pairwise network comparison between the 17 cancer networks, computed the $H_{JS}$ distances, and ranked the $H_{JS}$ distances from the smallest to the largest. Table 3 lists the results for the top three most similar pairs of cancer networks based on the $H_{JS}$ distance. It is noted that the $H_{JS}$ distance is non-zero, and the value may be as small as 0.0214. In other words, no two networks have exactly the same subgraph frequency distributions. The complete list of the $H_{JS}$ distances of the cancer networks is given in File S7.

The degree of similarity between two networks is characterized by subgraph frequency distributions. Given a pair of highly similar networks, we calculated the absolute values of the difference of the normalized frequency distributions of the network subgraphs. The magnitude of the difference (both three-node and four-node subgraphs) can be seen in the right-hand side of Table 3. The range of difference of the three-node subgraph distributions for the "acute myeloid leukemia (AML)" network and the "pancreatic cancer" network is
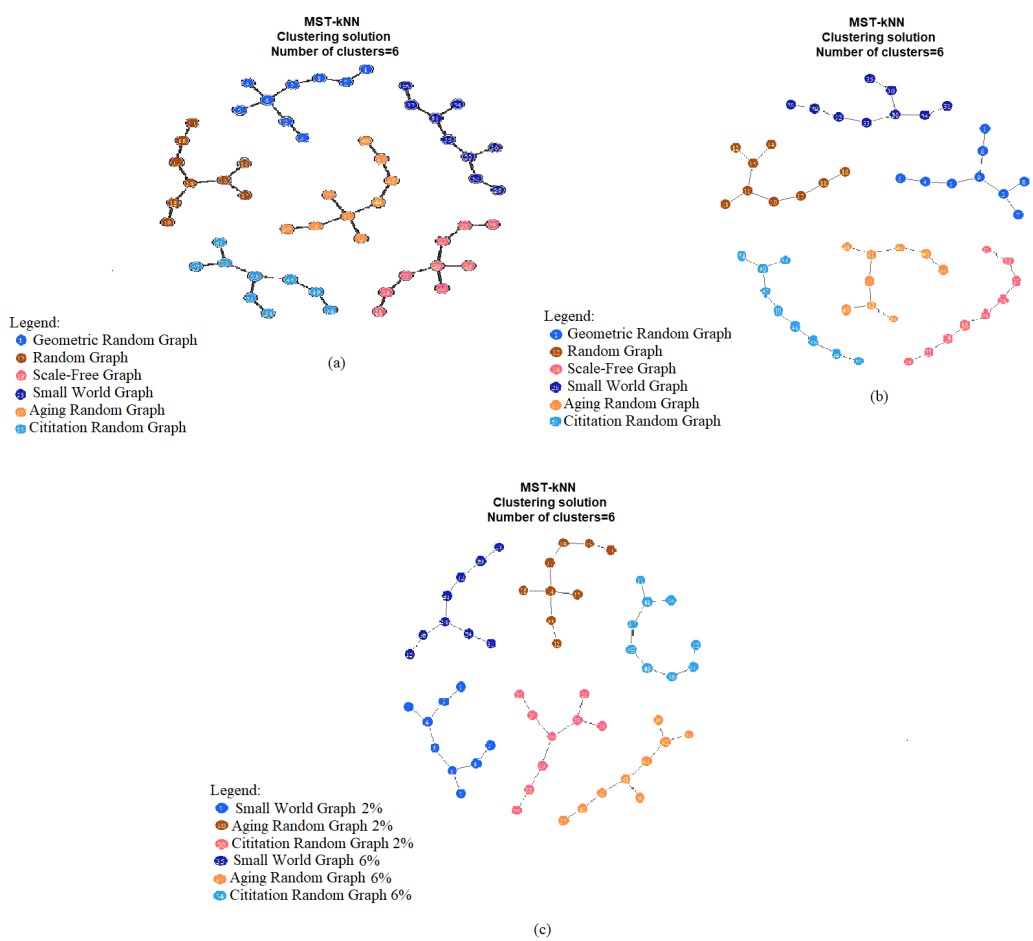

**Figure 1** Visualization of the results of classification for (A) simulation I, (B) simulation II and (C) simulation III, using MST-kNN.

**Table 3** The top three most similar pairs of cancer networks based on the $H_{JS}$ distance of the three-node subgraphs and four-node subgraphs normalized frequency distributions.

| Network A | Network B | $H_{JS}$ distance | Absolute value of the difference of the subgraph normalized frequency distributions |
|---|---|---|---|
| | 3-node subgraph ID | | (ID 6, ID 12, ID 36, ID 38) |
| Acute myeloid leukaemia (AML) | Pancreatic cancer | 0.0214 | (0.013, 0.007, 0.021, 0.002) |
| Chronic myeloid leukaemia (CML) | Gastric cancer | 0.0309 | (0.038, 0.020, 0.018, null) |
| Gastric cancer | Small cell lung cancer | 0.0339 | (0.024, 0.042, null, 0.018) |
| | 4-node subgraph ID | | (ID 14, ID 28, ID 74, ID 76, ID 78, ID 280, ID 328, ID 392, ID 2184) |
| Gastric cancer | Hepatocellular carcinoma | 0.0998 | (0.001, 0.027, 0.063, 0.011, null, 0.001, 0.045, 0.046, 0.006) |
| Chronic myeloid leukaemia (CML) | Melanoma | 0.138 | (0.009, 0.049, 0.114, 0.017, null, 0.028, 0.016, 0.016, null) |
| Hepatocellular carcinoma | Small cell lung cancer | 0.138 | (0.115, 0.006, 0.025, 0.015, null, 0.004, 0.004, 0.065, 0.002) |

0.002–0.021, which is smaller than the ranges for the other two pairs of cancer networks. Both the "AML" and "pancreatic cancer" networks are characterized by the frequency distributions of four subgraphs: SIM (id_6), cascade (id_12), MIM (id_36), and FFL (id_38). The word "null" denotes that the subgraph normalized frequency distribution is zero. In our previous study (*Huang et al., 2020*), we have shown that subgraphs "id_6", "id_12", and "id_36" are not composed of any three-node subgraphs. We considered that these three subgraphs exhibit the property of *irreducibility* (a N-node subgraph does not embed any other N-node subgraph) (*Huang et al., 2020*).

The range of difference of the four-node subgraph distributions for "gastric cancer" and "hepatocellular carcinoma" is 0.001–0.0046, which is lower than those of the other two pairs of cancer networks.

We formally define the *irreducibility* property of a graph as follows: for the set of $N$-node directed graph, graph $G$ is said to be irreducible if graph $G$ has exactly $N$-1 links. It is easy to see that the graph with irreducibility property is the basic block for constructing the $N$-node structure motifs. For the four-node subgraphs: "id_14" (SIM), "id_28", "id_74", "id_76" (MIM), "id_280", "id_328" (cascade), "id_392", and "id_2184", we considered these eight subgraphs also exhibit the property of irreducibility (*Huang et al., 2020*). In fact, we found that the underlying substructures of molecular networks are dominated by irreducible subgraphs, and accordingly, this behavior also holds true for the STN and cellular process networks. According to our previous work (*Huang et al., 2020*), the 'reciprocity' of these 'irreducible building blocks' are all negative. Reciprocity is a parameter that quantifies the degree of bidirectional connection of a network subgraph. Molecular networks are mostly composed of these types of subgraphs, a possible reason is that the signal can be quickly transmitted from the cell membrane to the nucleus, there is no feedback signal. Furthermore, since the irreducible subgraphs are the dominate subgraphs of molecular networks, one can apply the dimension reduction technique to reduce the complexity of large networks, while preserving the algorithmic information content the networks very well (*Zenil, Kiani & Tegnér, 2015*; *Kiani et al., 2016*).

Given the three-node subgraph normalized frequency distributions of the 17 cancer networks, the AML and pancreatic cancer networks (hsa05221 and hsa05212) exhibited the smallest $H_{JS}$ distance. We used the "User data mapping" application provided by the KEGG resource, and accordingly, the regulatory relations among the genes embedded in the three-node subgraph module (blue color) are depicted in Figs. 2A and 2B. According to the KEGG annotation, both networks involve six common biological processes. These six processes are located in three different regions in Figs. 2A and 2B. The upper part of Figs. 2A and 2B refers to the PI3K-Akt, MAPK, and Jak-STAT signaling processes; the right-hand part consists of the apoptosis and proliferation process; and the lower part includes the cell cycle process. Thus, our findings suggest that the underlying signaling mechanisms and cellular processes associated with the two networks are highly similar. These results are unexpected because the identities of the genetic elements are not considered in our calculations, and the results are inferred only from the subgraph frequency distributions. This study takes an important step in the direction of defining the relationship between subgraph distributions and subgraph-associated signal transduction and/or cellular processes. For instance, for

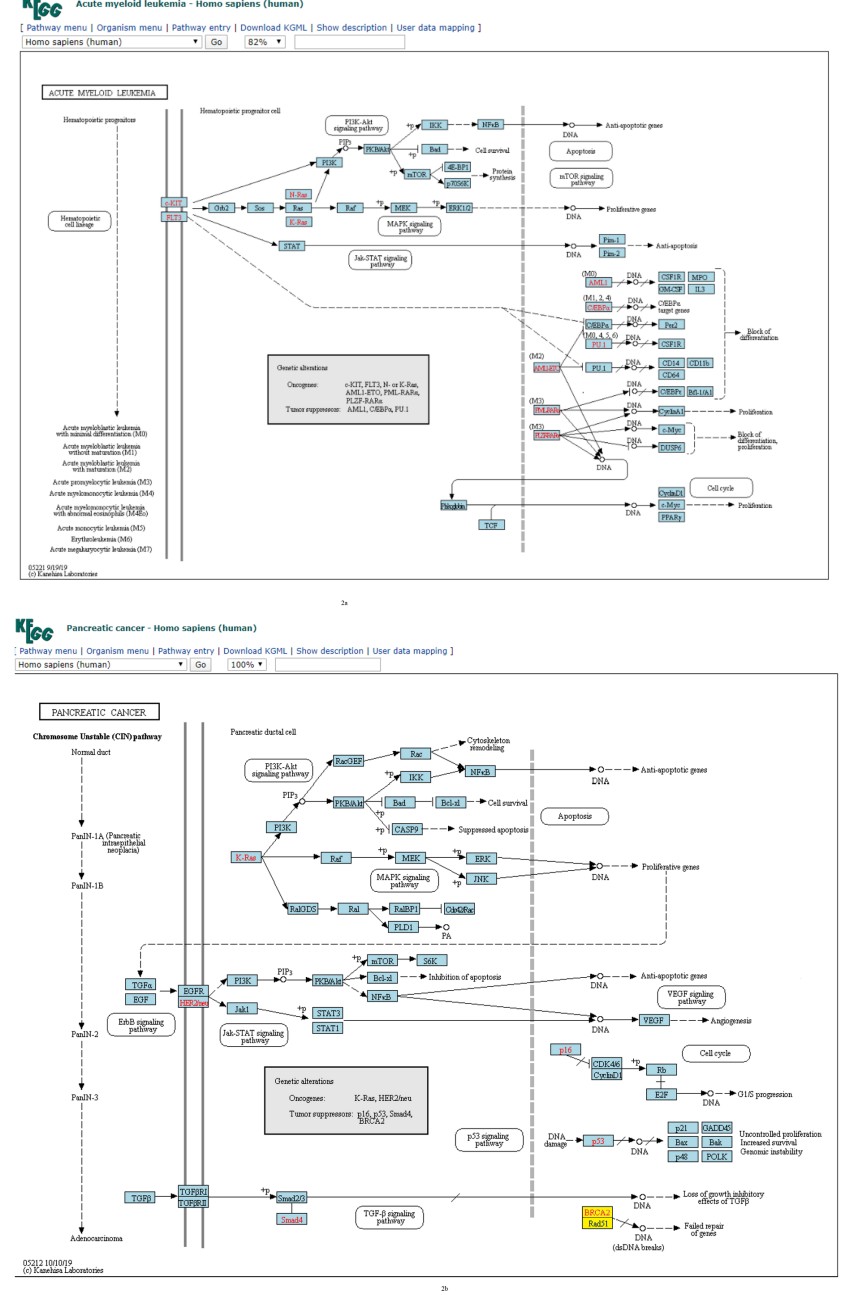

**Figure 2** **(A) Three-node subgraph module of the AML network. (B) Three-node subgraph module of the pancreatic cancer network.** Blue color boxes denote genes embedded in a subgraph module. Other colored objects denote genes which are not belong to a subgraph module, red color fonts mean genetic alternation (oncogene or tumor suppressor gene).

both AML and pancreatic cancer, the Ras-PI3K-PKB/Akt subgraph and Ras-Raf-MEK subgraph are associated with the PI3K-Akt and MAPK signal transduction pathways, respectively.

Furthermore, similar observations were found when considering four-node subgraph frequency distributions. Genes embedded in the four-node subgraph modules of the gastric cancer and hepatocellular carcinoma networks are depicted in Figs. 3A and 3B, respectively. Genes embedded in the four-node subgraph modules are shown in blue. It is clear that both cancer networks are mainly composed of the "SIM" pattern (id_14) and the "id_280" pattern (id_280). As shown in Table 3, the $H_{JS}$ distance of id_6 is as small as 0.001. According to the annotation provided by KEGG, both cancer networks involve six common processes. The left-hand part consists of the Wnt, PI3K-Akt, TGF-$\beta$, and MAPK signaling processes, and the right-hand part consists of the cell cycle and proliferation processes.

For four-node subgraphs, the above figures shown that we have identified the scaffold (the annotation provided by KEGG), it is a set of physically bound proteins which maintain the specificity of the signal transduction pathway and catalyze the activation of the pathway components (*Burack & Shaw, 2000*). In a review article, Koch reported out that the forkhead box family transcription factors may affect the Wnt signaling activity that leads to various types of cancer (*Koch, 2021*), and Wnt pathway is a drug therapeutic target (*Krishnamurthy & Kurzrock, 2018*; *Jung & Park, 2020*). The above findings provide an example of biological application of our approach.

In Fig. 4A, we plotted the normalized frequency of the three-node subgraphs for the three pairs of cancer networks. The AML network and the "pancreatic cancer" network were found to be highly similar because the $H_{JS}$ distance is the smallest among all possible pairwise comparisons. Figure 4A depicts that the red and green dots are located close to each other for the following four subgraphs: SIM (id_6), cascade (id_12), MIM (id_36), and FFL (id_38); hence, the $H_{JS}$ distance between the AML and pancreatic cancer networks is minimal.

"Chronic myeloid leukemia (CML)" and "gastric cancer" are associated with the second smallest $H_{JS}$ distance. These two networks are characterized by similar frequency distributions of the following subgraphs: SIM, cascade, and MIM (Fig. 4A, purple and blue dots).

## Biological interpretation—cancer networks

Regarding the practicality of our approach, we show that our method is able to cluster cancer networks with similar underlying regulatory topology. It was found that the AML and pancreatic cancer networks exhibited the smallest $H_{JS}$ distance. AML and pancreatic disease have been reported to be observed simultaneously in some patients during clinical diagnosis (*Cascetta et al., 2014*). Pancreatic masses develop during or after AML (*Messager et al., 2012*). AML can rarely mimic the clinical picture of pancreatic cancer (*Schafer et al., 2008*), while pancreatitis is a characteristic in the appearance of AML (*De Castro, Vencer & Espinosa, 2017*). According to the KEGG annotation, both cancer networks involve three common signaling pathways; that is. PI3K-Akt, MAPK and Jak-SAT. For instance, given

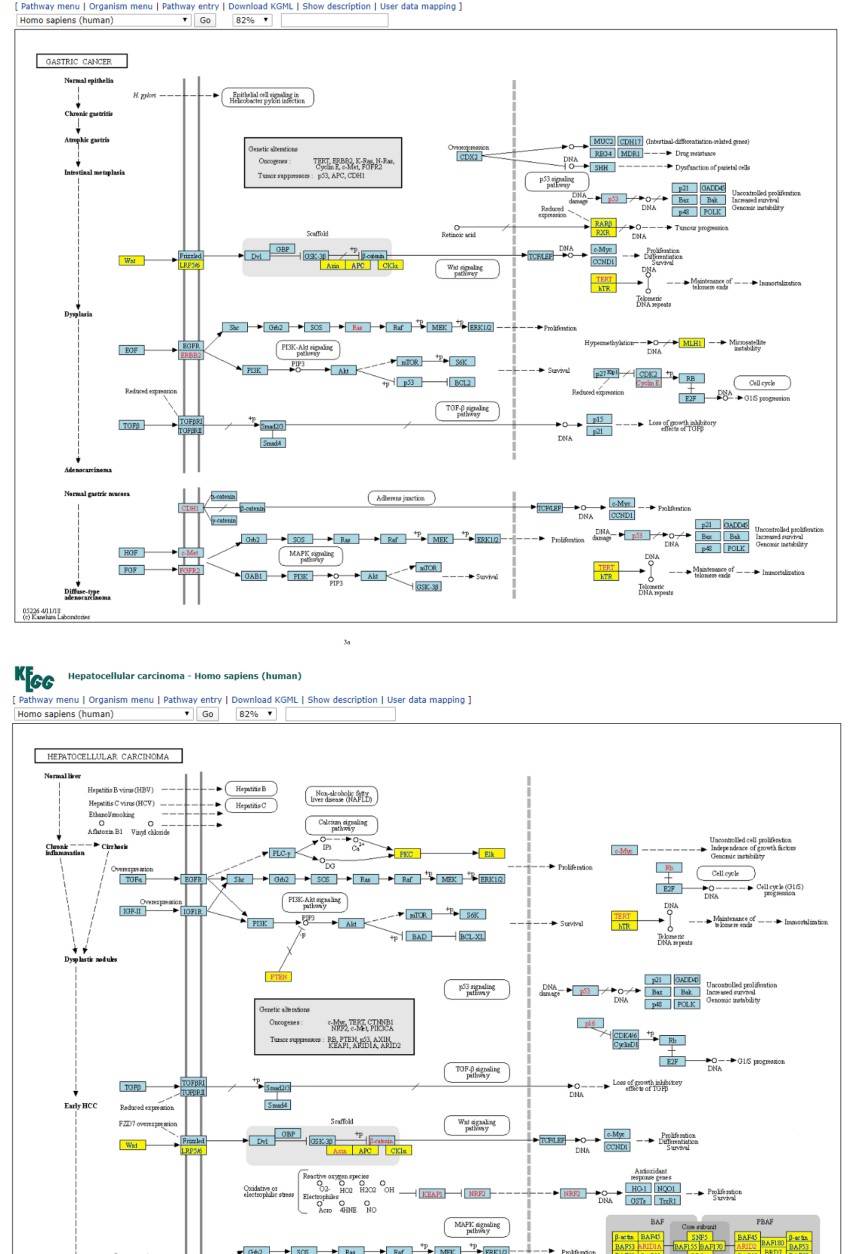

**Figure 3 (A) Four-node subgraph module of the gastric cancer network. (B) Four-node subgraph module of the hepatocellular carcinoma network.** Blue color boxes denote genes embedded in a subgraph module. Other colored objects denote genes which are not belong to a subgraph module, red color labels mean genetic alternation (oncogene or tumor suppressor gene), and grey colored rectangle in the middle of Figs. 3A and 3B denote scaffold (KEGG annotation).

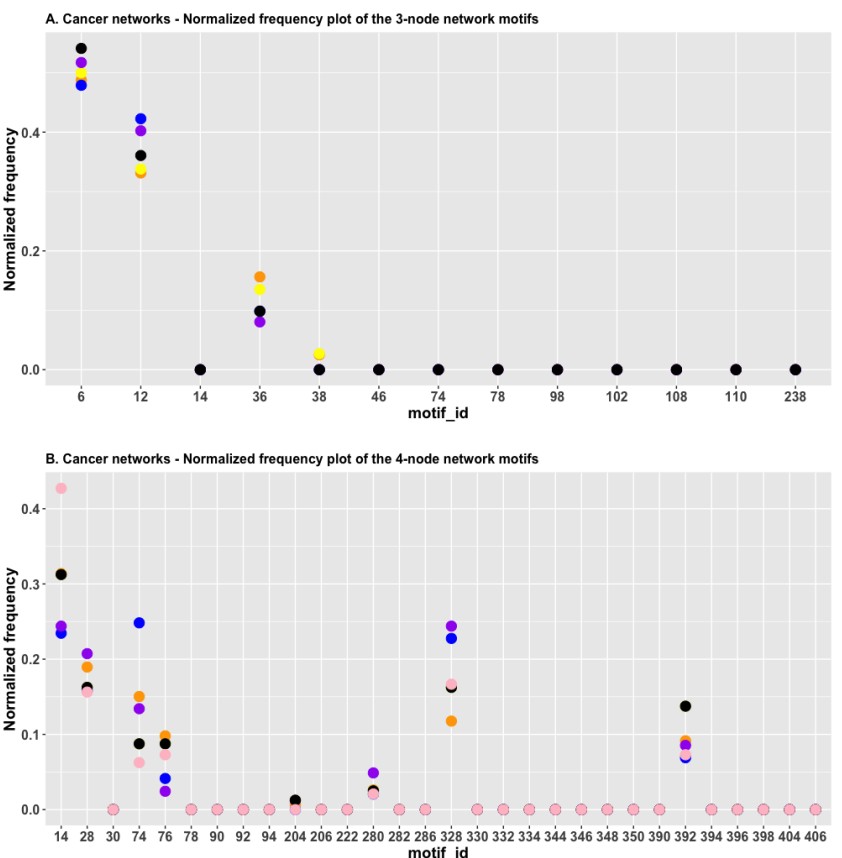

**Figure 4** (A) The plot of the normalized frequency of the three-node subgraphs for the three pairs of cancer networks with the smallest $H_{JS}$ distance. Color labelling of the cancer types: AML (orange), pancreatic cancer (yellow), CML (blue), gastric cancer (purple) and small cell lung cancer (black). (B) The plot of the normalized frequency of the four-node subgraphs for the three pairs of cancer networks with the smallest $H_{JS}$ distance. Color labelling of the cancer types: Gastric cancer (orange), Hepatocellular carcinoma (yellow), Chronic myeloid leukaemia (blue), Melanoma (purple) and Small cell lung cancer (black).

the common PI3K-Akt pathway found in both diseases, we found the following two sets of common activation relationships: (i) Ras → PI3K → PKB/Akt → IKK → NFkB, and (ii) Ras → raf → MEK → ERK. Thus, the results suggest that the underlying signaling mechanisms associated with the two cancer diseases are highly similar. Even though some of the genes are different in the two diseases, the same signaling mechanisms are involved; hence, the subgraph-based approach take us from the subgraph level to the mechanism level interpretation.

For the second pair of networks, previous studies have shown that (i) both CML and gastric cancer might co-exist in a single patient (*Butala, Kalra & Rosner, 1989*), which might be due to decreased immunity (*Mangal et al., 2018*), and (ii) clinicians are recommended to pay attention to the association of CML and gastric cancer (*Mokhtarifard et al., 2016*) and expression of *MMP1* may contribute to gastric cancer formation (*Yang et al., 2017*).

For the third pair of networks, it has been reported that (i) gastrointestinal metastases from lung cancer are diagnosed at a late stage and are thus life-threatening (*Li et al., 2018*); (ii) gastric cancer metastasis may result from lung cancer (*Gao et al., 2015*); (iii) gastric cancer is associated with lung cancer (*Park, 1998*; *Snyder et al., 2013*; *Nitipir et al., 2018*; *Koh & Lee, 2014*) ; (iv) both diseases can be successfully treated with carboplatin (*Sano et al., 1986*). The use of natural products along with chemotherapy drugs are effective in treating lung cancer  (*Huang et al., 2017*).

Figure 4B depicts the plot of the normalized frequency distributions of the four-node subgraphs for cancer networks. The normalized frequency counts of most of the four-node subgraph patterns are zero; therefore, only the IDs of the first 30 subgraphs (sorted according to the decimal representation) are shown. The difference between the distributions in gastric cancer (red) and hepatocellular carcinoma (HCC) (green) is minimal among the following subgraphs: id_14, id_28, id_76, id_204, and id_280, whereas the difference between CML (blue) and melanoma (purple) is minimal among a different set of subgraphs: id_14, id_76, id_204, and id_328.

According to many studies, (i) both gastric cancer and HCC can co-exist in the same patient (*Sakumura, Tajiri & Sugiyama, 2018*; *Hu et al., 2014*); (ii) optimal surgical strategies have been developed for treating synchronous gastric cancer and HCC (*Tawada et al., 2014*; *Uenishi et al., 2003*); and (iii) it is necessary to closely follow up patients with gastric cancer or HCC for an early diagnosis (*Chen et al., 2017*). Next, we used the R package to perform cluster analysis of the cancer networks based on the $H_{JS}$ distance measure. In Figs. 5A and 5B, we show the heatmap of the cancer networks using the $H_{JS}$ distance of the three-node subgraphs and the four-subgraph nodes, respectively. For the three-node subgraphs, we identified the following three pairs of highly similar networks: (i) "acute myeloid leukemia" and "pancreatic cancer", (ii) "gastric cancer" and "small cell lung cancer", and (iii) "chronic myeloid leukemia" and "hepatocellular carcinoma". For the four-node subgraph clustering results, the following highly similar pairs are detected: (i) "gastric cancer" and "hepatocellular carcinoma", (ii) "chronic myeloid leukemia" and "melanoma", and (iii) "basal cell carcinoma" and "small cell lung cancer". In the majority, the results of the identified pairs of networks are consistent with the findings listed in Table 3.

From Fig. 5A, we noted two regions associated with a similar color, *i.e.*, the upper left-hand and lower right-hand corners. Similar patterns can be found for the four-node subgraphs, *i.e.*, Fig. 5B. The result suggests a group of cancer networks possess similar subgraph topology.

## Signal transduction network (STN)

A list of the top three most similar STN is given in Table 4. The range of difference for the three-node subgraph distributions of the first pair of highly similar networks, "sphingolipid signaling pathway" and "TGF-beta signaling pathway", is 0.001–0.029, which is relatively small when compared to the other two pairs of pathways. The range of difference for the four-node subgraph distributions of the first pair of highly similar networks is 0.001–0.034, which is lower than that of the other two pairs of the STN. Again, it was found that the

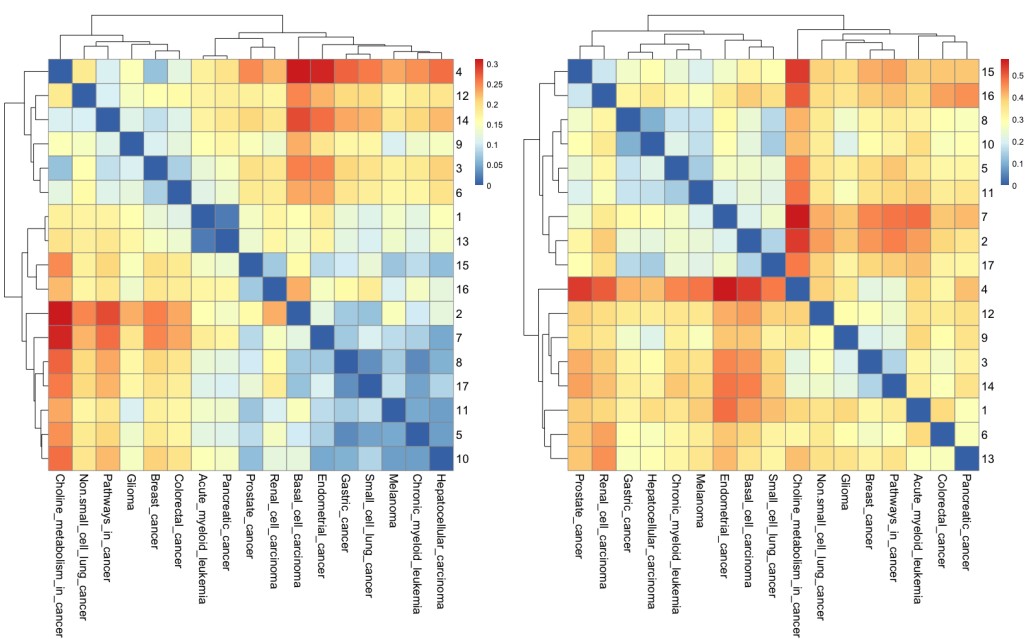

**Figure 5** The heatmap of $H_{js}$ distance computed for cancer networks: (A) three-node subgraphs and (B) four-node subgraphs, with light yellow and darker red denote the small and large value of $H_{js}$ respectively. The number of each row denotes the network name (File S1).

**Table 4** The top three most similar pairs of STN based on the $H_{JS}$ distance of the three-node subgraphs and four-node subgraphs normalized frequency distributions.

| Network A | Network B | $H_{JS}$ distance | Absolute value of the difference of the subgraph normalized frequency distributions |
|---|---|---|---|
| | 3-node subgraph ID | | (ID 6, ID 12, ID 36, ID 38) |
| Sphingolipid signaling pathway | TGF-beta signaling pathway | 0.0263 | (0.001, 0.029, 0.026, 0.005) |
| ErbB signaling pathway | Hippo signaling pathway | 0.0312 | (0.015, 0.017, 0.037, 0.005) |
| PI3K-Akt signaling pathway | Ras signaling pathway | 0.0330 | (0.010, 0.010, 0.007, 0.007) |
| | 4-node subgraph ID | | (ID 14, ID 28, ID 74, ID 76, ID 78, ID 280, ID 328, ID 392, ID 2184) |
| Neurotrophin signaling pathway | Ras signaling pathway | 0.115 | (0.034, 0.030, 0.033, 0.026, 0.018, 0.024, 0.019, 0.008, 0.001) |
| Adipocytokine signaling pathway | B-cell receptor signaling pathway | 0.119 | (0.017, 0.054, 0.035, 0.034, 0.001, 0.013, 0.012, 0.031, 0.007) |
| Apelin signaling pathway | Chemokine signaling pathway | 0.131 | (0.045, 0.037, 0.011, 0.016, 0.005, 0.029, 0.010, 0.016, 0.007) |

underlying substructure of the molecular networks is dominated by irreducible subgraphs. The complete list of the $H_{JS}$ distances of the STN is given in File S7.

Figure 6A depicts the normalized frequency distributions of the three-node subgraphs for STN. Figure clearly shows that the red and green dots are located close to each other for the following four subgraphs: id_6, id_12, id_36, and id_38. This indicates that the $H_{JS}$ distance between the "sphingolipid signaling pathway" and "TGF-beta signaling pathway"

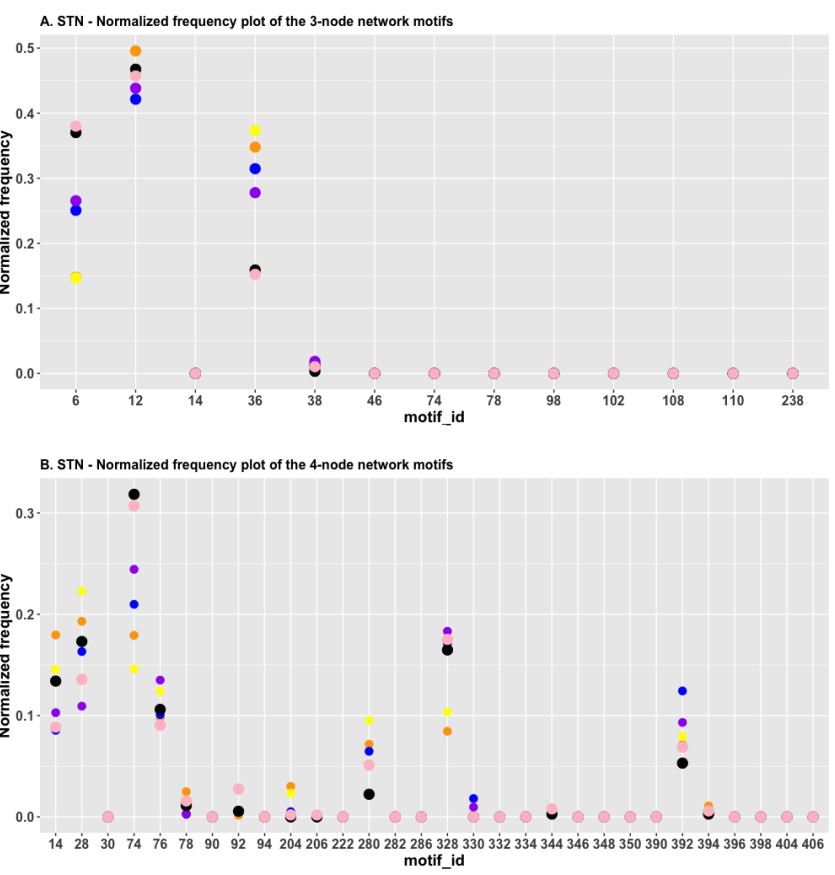

**Figure 6** **(A) The plot of the normalized frequency of the three-node subgraphs for three pairs of STN with the smallest $H_{JS}$ distance.** Color labelling of the STN is: Sphingolipid signaling pathway (orange), TGF-beta signaling pathway (yellow), ErbB signaling pathway (blue), Hippo signaling pathway (purple), PI3K-Akt signaling pathway (black) and Ras signaling pathway (pink). (B) The plot of the normalized frequency of the four-node subgraphs for the three pairs of STN with the smallest $H_{JS}$ distance. Color labelling of the STN are: Sphingolipid signaling pathway (orange), Ras signaling pathway (yellow), Adipocytokine signaling pathway (blue), B-cell receptor signaling pathway (purple), Apelin signaling pathway black) and Chemokine signaling pathway (pink).

is minimal. The blue and purple dots are located close to each other for the same set of subgraphs, suggesting that the $H_{JS}$ distance between the "ErbB signaling pathway" and the "Hippo signaling pathway" is minimal. Similarly, the black and pink dots are located close to each other, implying a low $H_{JS}$ distance between the "Apelin signaling pathway" and the "chemokine signaling pathway".

## Biological interpretation—STN

For the STN three-node subgraph case, the most similar pair of networks is the "sphingolipid signaling pathway" and the "TGF-beta signaling pathway". *Dennler, Goumans & Ten Dijke (2002)* demonstrated that endogenous sphingolipid mediators are involved in regulating the TGF-beta signaling pathway. This is further supported by two other studies, which demonstrated cooperation between TGF-beta and S1P

Peer J

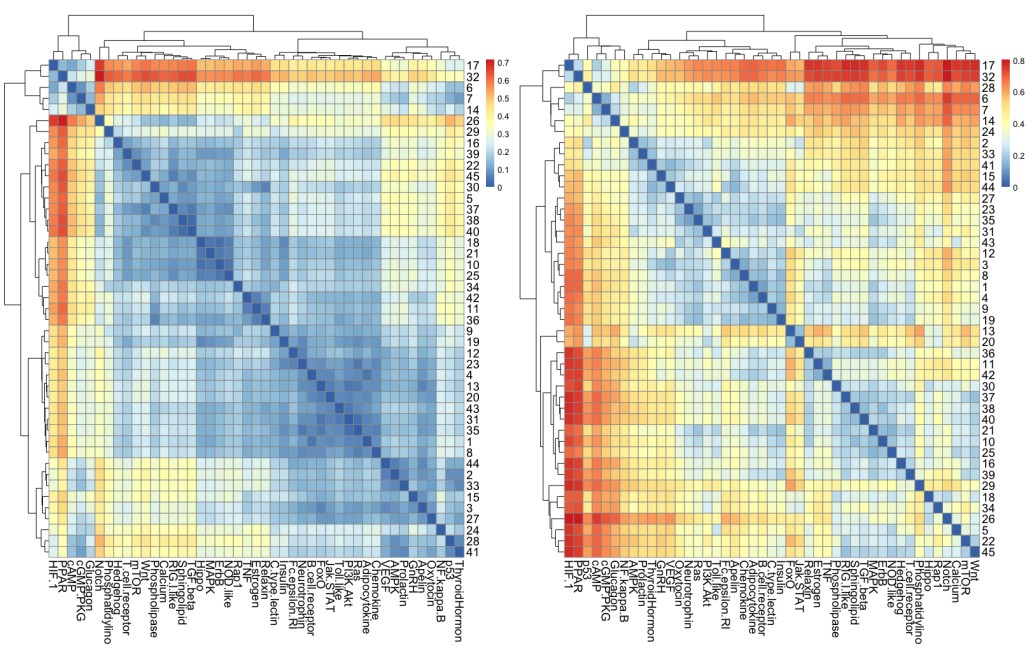

**Figure 7  The heatmap of $H_{JS}$ distance computed for STN: (A) three-node subgraphs and (B) four-node subgraphs, with blue and red denote the small and large value of $H_{JS}$, respectively. The number of each row denotes the network name (File S1).**

signaling (*Yamanaka et al., 2004*; *Xin et al., 2004*). In addition, owing to the cross-talk of these two pathways, novel, non-invasive therapies can be developed (*Nicholas et al., 2017*).

For the second pair of networks, both ErbB and Hippo signaling pathways are regulated by circular RNA and microRNA in hypopharyngeal cancer (*Feng et al., 2019*). For the third pair of networks, it was demonstrated that the testis-specific protein Y-linked 1 (TSPY1) activates the PI3K/AKT and RAS signaling pathways by suppressing IGFBP3 expression in lung adenocarcinoma and liver hepatocellular carcinoma progression (*Tu et al., 2019*). By inhibiting the EGFR/AKT pathway in oral cancer, quercetin appears to be an effective anti-tumor agent (*Chan et al., 2016*).

Figure 6B depicts the plot of the normalized frequency distributions of the 4-node subgraphs. Again, only the first 30 patterns are shown. The difference between the "neurotrophin signaling pathway" and the "Ras signaling pathway" (green) distribution is minimal among the following subgraphs: id_78 and id_204, whereas the difference between the "adipocytokine signaling pathway" (blue) and "B-cell receptor signaling pathway" (purple) is minimal among the following subgraphs: id_78, id_204, id_328, id_330, and id_2184 (not shown in Fig. 6B).

The results of cluster analysis for STN are given in Figs. 7A and 7B.

From Figs. 7A and 7B, we noted that certain areas are associated with similar colors. The result suggests a group of STN possess similar subgraph topology.

**Table 5  The top three most similar pairs of cellular processes based on the $H_{JS}$ distance of the three-node subgraphs and four-node subgraphs.**

| Network A | Network B | $H_{JS}$ distance | Absolute value of the difference of the subgraph normalized frequency distributions |
|---|---|---|---|
| | 3-node subgraph ID | | (ID 6, ID 12, ID 36, ID 38) |
| Cell-cycle | Cellular senescence | 0.0697 | (0.015, 0.078, 0.090, 0.004) |
| Apoptosis | Focal adhesion | 0.0839 | (0.095, 0.037, 0.041, 0.015) |
| Cellular senescence | Focal adhesion | 0.106 | (0.059, 0.098, 0.002, 0.034) |
| | 4-node subgraph ID | | (ID 14, ID 28, ID 74, ID 76, ID 78, ID 280, ID 328, ID 392, ID 2184) |
| Cell-cycle | Cellular senescence | 0.125 | (0.003, 0.0002, 0.001, 0.027, 0.001, 0.004, 0.057, 0.035, 0.039) |
| Apoptosis | Focal adhesion | 0.143 | (0.048, 0.013, 0.005, 0.041, 0.004, 0.023, 0.032, 0.016, 0.047) |
| Cell-cycle | Focal adhesion | 0.205 | (0.034, 0.047, 0.035, 0.008, 0.010, 0.083, 0.028, 0.054, 0.024) |

## Cellular processes

A list of the top three most similar $H_{JS}$ distances for cellular processes is given in Table 5. The range of difference of the three-node subgraph distributions for the "cell cycle" and "cellular senescence" networks is 0.004–0.090, which is lower than that of the other two pairs of networks. The range of difference of the four-node subgraph distributions for the "cell cycle" and "cellular senescence" networks is 0.0002–0.0057, which is smaller than that of the other two pairs of networks. Once again, the results suggested that the underlying modular structure of molecular networks is dominated by irreducible subgraphs. The complete list of the $H_{JS}$ distances of the cellular processes is given in File S7.

It is obvious that in Fig. 8A, the red and green dots are located close to each other for id_6 and id_38; this indicated that the $H_{JS}$ distance between the "cell cycle" and the "cellular senescence" processes is minimal. The blue and purple dots are located close to each other for FFL (id_38), which suggests that the $H_{JS}$ distance between the "apoptosis" and "focal adhesion" processes is small.

## Biological interpretation—cellular processes

For the three-node case, the cellular senescence phenomenon (*Sun, Coppe & Lam, 2018*) is highly relevant to the cell-cycle process. Most of the drugs with anti-cancer potential are inducing cell-cycle arrest (*Hsu & Chung, 2012*; *Czarnomysy et al., 2018*) and apoptosis (*Lee et al., 2017*; *Liu et al., 2019*). Cellular senescence is a phenomenon that is characterized by irreversible cell-cycle arrest (*Sun, Coppe & Lam, 2018*), and it results from the coordination of cell-cycle arrest and cell expansion (*Ogrodnik et al., 2019*).

For the second pair of networks, numerous studies have shown the regulatory relationship between focal adhesion and apoptosis. *Luo et al. (2018)* studied the effect of a green tea compound on the proliferation and apoptosis of breast cancer cells by inhibiting the focal adhesion kinase (FAK) signaling pathway. FAK is an important component in regulating endothelial cell apoptosis (*Suhr & Bloch, 2012*). Cance and Golubovskaya

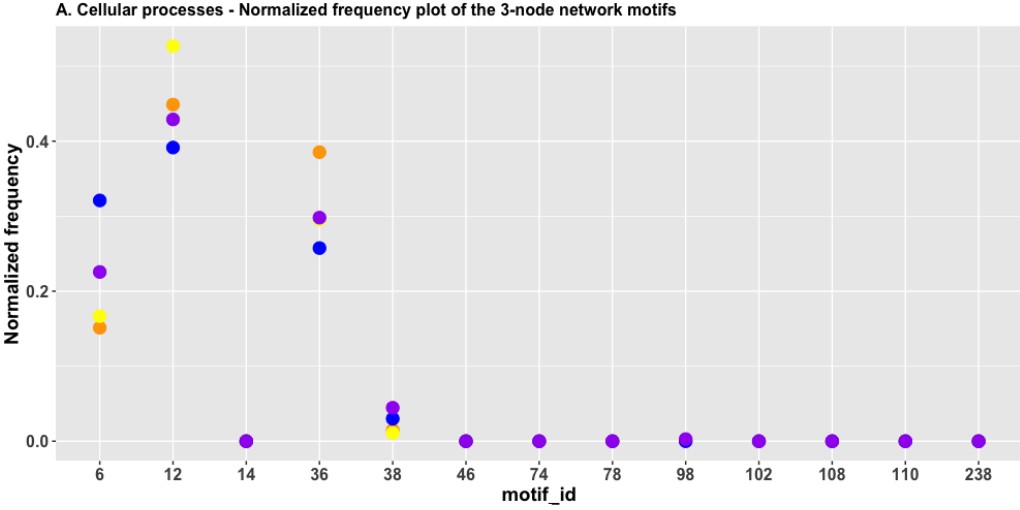

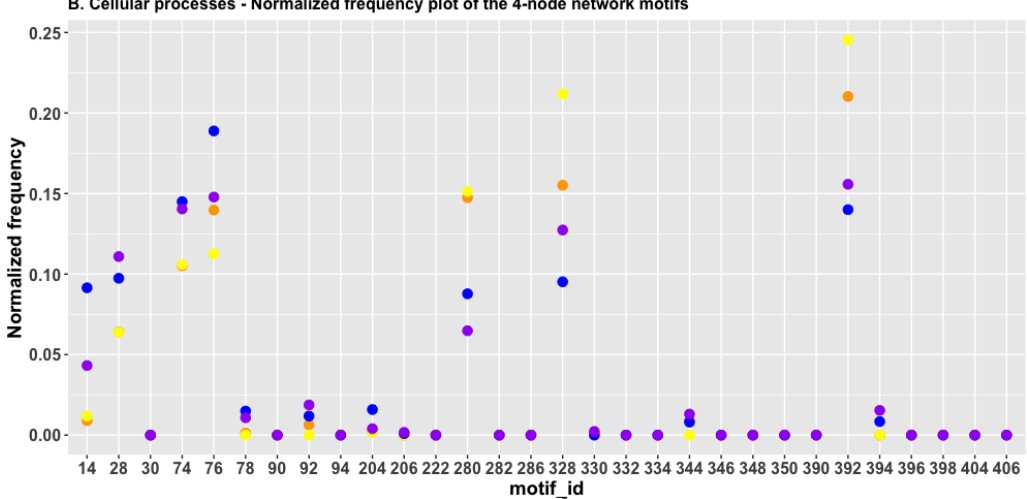

**Figure 8** (A) The plot of the normalized frequency of the three-node subgraphs for three pairs of cellular processes with the smallest $H_{JS}$ distance. Color labelling of the cellular processes: cell cycle (orange), cellular senescence (yellow), apoptosis (blue), and focal adhesion (purple). (B) The plot of the normalized frequency distributions of the four-node subgraphs for three pairs of cellular processes with the smallest $H_{JS}$ distance. Color labelling of the cellular processes are: cell cycle (orange), cellular senescence (yellow), apoptosis (blue) and focal adhesion (purple).

reported that the interaction of FAK and p53 may promote cell survival or induce cell apoptosis (*Cance & Golubovskaya, 2008*).

For the third pair of networks, it was found that caveolin-1 suppresses FAK activity and triggers morphological alterations of the cell, changes enzyme activities and gene expression (*Park, 2017*; *Cho et al., 2004*), and that the inhibition of FAK expression can activate the cellular senescence program (*Chuang et al., 2019*).

Figure 8B depicts the plot of the normalized frequency distributions of the four-node subgraphs for cellular processes. Again, only the first 30 patterns are shown. The difference between the "cell cycle" (red) and "cellular senescence" (green) distributions is minimal
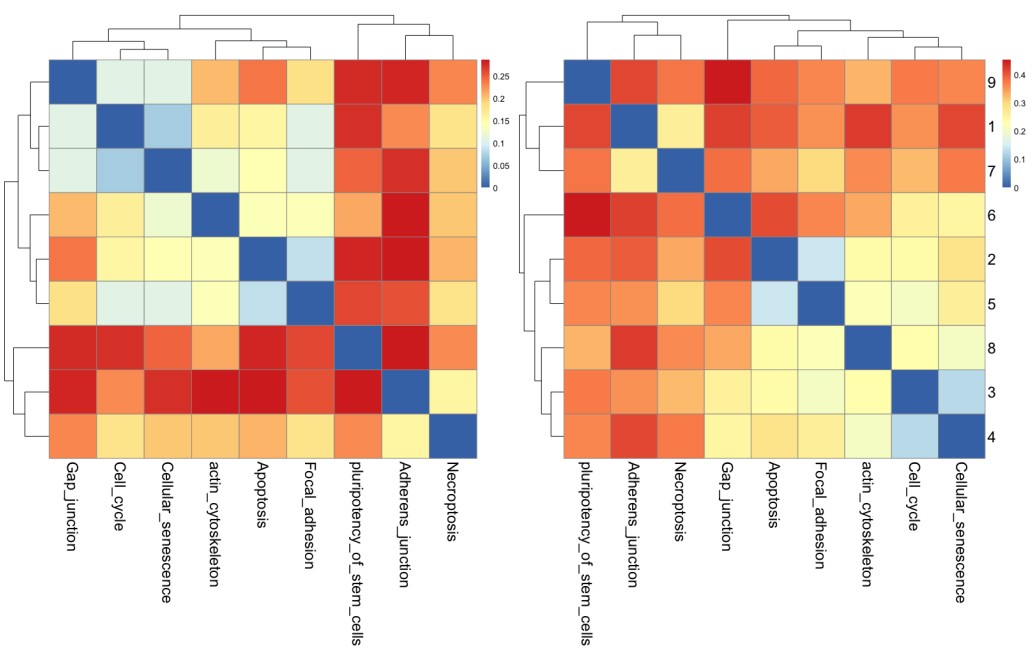

**Figure 9** **The heatmap of $H_{JS}$ distance computed for cellular processes: (a) three-node subgraphs and (b) four-node subgraphs, with light yellow and darker red denote a small and large value of $H_{JS}$ respectively.** The number of each row denotes the network name (File S1).

among the following subgraphs: id_14, id_28, id_74, id_78, id_204 (0.001, not shown in Table 5), and id_280, whereas the difference between "apoptosis" (blue) and "focal adhesion" (purple) is minimal among the following subgraphs: id_74, id_78, id_92, id_206, id_330, and id_344 (the last four IDs are not shown in Table 5, and the normalized frequency magnitudes associated with them are small). For the third pair, the difference between "cell cycle" (red) and "focal adhesion" (purple) is minimal among the following subgraphs: id_76, id_78, id_204, id_206, id_330, and id_344 (the last four IDs are not shown in Table 5, and the normalized frequency magnitudes associated with them are small).

The results of cluster analysis for the cellular processes are given in Figs. 9A and 9B.

From Fig. 9A, the three-node subgraph clustering results, we identified the following pairs of highly similar networks: (i) "cell cycle" and "cellular senescence" and (ii) "apoptosis" and "focal adhesion". The results for the identified pairs of networks are consistent with the findings in Table 5.

For the four-node subgraphs, i.e., Fig. 9B, the following highly similar pairs are detected: (i) "cell cycle" and "cellular senescence" and (ii) "apoptosis" and "focal adhesion". Moreover, the results of the identified pairs of networks are consistent with the findings listed in Table 5.

We also studied the inter-quartile range and median value of the $H_{JS}$ distances of the cancer networks, STN and cellular processes; the results are given in File S8.

## CONCLUSIONS

We studied the problem of determining the similarity of two directed networks by proposing an effective approach, the subgraph-based approach. First, the normalized frequency distributions of the three-node subgraphs and four-node subgraphs for three major types of molecular networks were calculated by using our algorithm, *PatternFinder*. Compared to other algorithms, *i.e. LoTo* and *acc-Motif*; our subgraph detection algorithm obtained similar results but had certain advantages.

Second, we conducted three simulation experiments, which confirmed the superiority of our method. The simulation experiments considered six network models along with three different network sizes and two edge densities. The accuracy of classification based on subgraph-based approach with information-theoretic entropy ($H_{JS}$) is 100%, which is better than the current status of other works.

Third, we used $H_{JS}$ to infer pairs of networks that exhibit similar/different regulatory interaction topologies. In particular, our results suggested that there are common regulation modules for AML and pancreatic cancer formation. To the best of our knowledge, the present study is the first to combine the network subgraph concept and $H_{JS}$ to address the molecular network similarity problem.

Fourth, we found that the underlying substructures of the molecular networks are dominated by irreducible subgraphs. This behavior holds true for the three classes of molecular networks we studied.

This study provides a systematic approach to dissect the underlying structures of molecular networks. We hypothesize that network structures can be understood in terms of the network subgraphs.

In future, the next step is to test our hypothesis by conducting five-node subgraphs analysis. As a first step towards five-node subgraph analysis, we have published papers on how to generate all the five-node subgraphs (*Efendi Zaenudin et al., 2019*). Regarding the applications of the five-node subgraph study, we plan to examine the association of the five-node subgraph modules and driver genes for cancer networks. Cancer driver genes are genes that give selective advantage for cancer progression. The level of association/enrichment is given by using odds ratio (OR). An OR >1 indicates that driver genes are enriched in the subgraph module. In our previous work (*Huang et al., 2020*), we examined the association of both the three-node and four-node subgraph modules and driver genes, and have found that many cancer networks, STN, cellular processes have an OR > 1. Similarity, we also have investigated the association of the subgraph module and essential genes (*Efendi, Huang & Ng, 2021*), but there are only a few networks enriched with essential genes. In conclusion, we have proposed a novel and effective approach, subgraph-based method, to compare and classify molecular networks with diverse functionalities.

## ACKNOWLEDGEMENTS

We would like to thank Mr. Ci-Jun Peng and Mr. I-Lun Hsieh, who spent efforts on developing the codes. We also thank the 'Editage Professional English Editing Service, Cactus Communications', for editing the English.

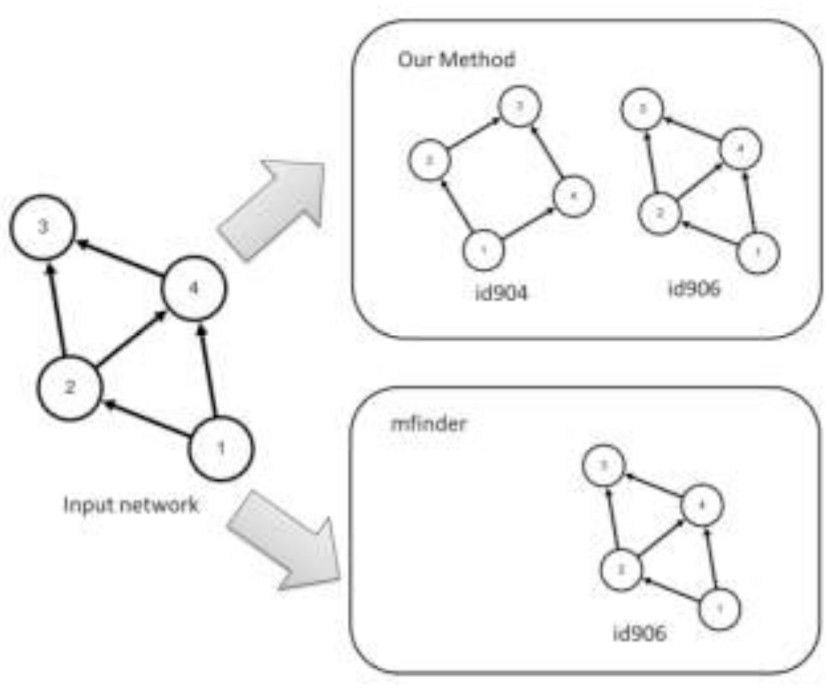

**Figure A1** **A comparison of two identification algorithms: *mfinder* and *PatternFinder*.**

## APPENDIX

### Subgraph identification tool—the *PatternFinder* algorithm

Many network motif detection tools have been developed to detect network motifs; including: FANMOD, MAVISTO, MFINDER, NetMatch, and SNAVI. We have reported (*Hsieh et al., 2015*) that those tools have at least two limitations: (i) motifs identified in one round may not be recoverable in another round because of the use of randomized algorithm, and (ii) nodes identities are missing.

We developed an algorithm named *PatternFinder* (*Huang et al., 2020*; *Lee, 2016*) to identify: (i) both three-node and four-node subgraphs in a network, and (ii) functional subgraphs embedded in the three-node subgraphs and four-node subgraphs not identified by MFINDER. We refer the reader to reference *Lee (2016)* for a more detail description of *PatternFinder*. Below we briefly summarized the *PatternFinder* algorithm.

Given the "Input network", *PatternFinder* is able to identify two four-node subgraphs, *i.e.*, the subgraph 'id_904' and subgraph 'id_906', whereas MFINDER can identify the subgraph 'id_906' only (Fig. A1). MFINDER considers motif 'id_904' is an independent subgraph. *PatternFinder* is able to identify subgraphs embedded in a subgraph.

In the following, a four-node subgraph is used as an example to illustrate the basic concept behind the *PatternFinder* algorithm. Given a network called '*net*' composes of 20 nodes, an adjacency matrix can be constructed. Let $n$ denotes the total number of nodes. Assuming that we want to identify a subgraph, denoted by 'id_2204', *PatternFinder* read in the '2204' pattern. This subgraph composes of four nodes and five edges, where the

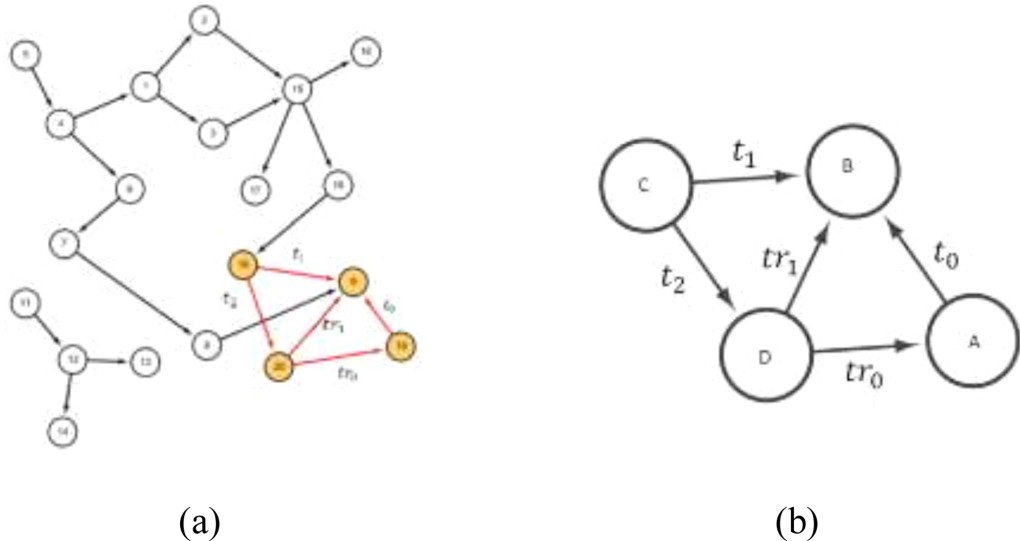

(a)                                     (b)

**Figure A2** **(A) An input network named '*net*' and (B) the four-node subgraph 'id_2204'.**

edges are denoted by $t_0$, $t_1$, $t_2$, $tr_0$ and $tr_1$. Starting from node A, *PatternFinder* begins to examine the following patterns: (i) is node A and node B connects with an edge $t_0$, (ii) is node B and node C connects with an edge $t_1$, (iii) is node C and node D connects with an edge $t_2$, and (iv) is node D and node A connects with an edge $t_0$, and node D and node B connects with an edge $tr_1$.

Starting from the network named '*net*', the algorithm begins the search from node 1 and labels it as node A. Node 1 and node 2 or node B are linked, the edge is denoted by *edge* (1, 2). The algorithm continues to search if there is a node links to node B, if not, the algorithm will relabel node B to node 3 and repeat the search. From Fig. A2, it was found that $A = 19$, $B = 9$, $C = 10$, and $D = 20$ are connected by three edges, *i.e.*, *edge* $(19,9) = t_0$, *edge* $(9,10) = t_1$, *edge* $(10,20) = t_2$, hence, four nodes are identified. However, according to the subgraph 'id_2204', there are two more edges which need to be determined, *i.e.*, *edge* $(20,9) = tr_1$ and *edge* $(20,19) = tr_2$. The computation time complexity of the algorithm *PatternFinder* is $O(n^4)$.

### Funding
The work of Chien-Hung Huang is supported by the grant of the Ministry of Science and Technology of Taiwan (MOST) (grant number MOST 109-2221-E-150-036). Dr. Ka-Lok Ng and Efendi Zaenudin works are supported by the MOST (grant numbers MOST 109-2221-E-468-013 and MOST 108-2221-E-468-020) and grants from Asia University (grant number ASIA-110-CMUH-12 ). The funders had no role in study design, data collection and analysis, decision to publish, or preparation of the manuscript.

### Grant Disclosures

The following grant information was disclosed by the authors:
Ministry of Science and Technology of Taiwan (MOST): MOST 109-2221-E-150-036.
MOST: MOST 109-2221-E-468-013, MOST 108-2221-E-468-020.

### Competing Interests

The authors declare there are no competing interests.

### Author Contributions

- Chien-Hung Huang and Ka-Lok Ng conceived and designed the experiments, performed the experiments, analyzed the data, prepared figures and/or tables, authored or reviewed drafts of the paper, and approved the final draft.
- Efendi Zaenudin performed the experiments, analyzed the data, prepared figures and/or tables, authored or reviewed drafts of the paper, and approved the final draft.
- Jeffrey J.P. Tsai conceived and designed the experiments, authored or reviewed drafts of the paper, managed the project and secured the necessary acquisition funding, and approved the final draft.
- Nilubon Kurubanjerdjit performed the experiments, prepared figures and/or tables, and approved the final draft.

### Data Availability

   The raw data is available in the Supplementary Files.

### Supplemental Information

Supplemental information for this article can be found online at http://dx.doi.org/10.7717/peerj.13137#supplemental-information.

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
