# Peer review of "Network subgraph-based approach for analyzing and comparing molecular networks"

_PeerJ, doi:10.7717/peerj.13137_

## Round 0.1 · original submission · Major Revisions

Overall, the paper was well received by reviewers, but all remark that certain areas of literature are not discussed, paper could be made more concise, and the details and results could be better presented. Notwithstanding reviewer 2's requests to include several papers from a single research group, please address all other reviewers' concerns, and in addition take account of my own recommendations below.

I have also examined your manuscript in detail, and whilst the results you present are convincing, the way that the manuscript is structured makes it very difficult to grasp exactly what contribution this work makes to the field. For instance, why is an alignment free method for measuring similarity better than an alignment based one ? Why are the (very many) general graph similarity metrics less suitable for this application ? Why is identification of 'irreducible building blocks' an advance of our understanding of biological signalling networks ? Providing a balanced introduction and discussion of these key points, and summarising your works contribution to them in the conclusion would help. There are also certain omissions with regard to code availability that must be addressed (simpy referencing other publications is not enough here).

I also have certain scientific questions:
Q1. Typically, biologists are interested in the 'significant' similarities between networks (for a range of reasons that should perhaps be apparent in this work). Doesn't introducing the H_js metric simply reformulate the question of 'significant similarity' between networks as a statistical analysis of H_js distributions, rather than statistics based on subgraph occurence ?

In your comparisons of 3-node and 4-node subgraph biological networks, you highlight the subgraphs present in both networks according to the metric in blue in each figure. I find it unsurprising that these common regions were identified given only the edge relationships, since both networks are drawn from the same underlying organismal network, and it is generally recognised that subgraphs of biological networks do have characteristic topologies, regardless of their edge or vertex labelling. Indeed - most approaches for determining common pathways in network rely on this property - and simply rank genes according to their enrichment in the observed graphs.
Q2. Is there a way of producing an 'alignment' between networks from their common subgraphs that would allow a more direct visualisation of the common regions ?
Q3. Are there situations where your label free approach would be superior to simple frequentist approaches such as occurence/common-edge counting ?
Q4. In Fig 15 and lines 427-435 you discuss the behaviour of H_js with respect to subgraph order (node number) and that H_js variance is greater for comparisons of STNs. This discussion seems anecdotal, and highlights an aspect of the H_js metric which is not properly examined: its variational properties. I would recommend downplaying this analysis and perhaps relocating it in its entirety to supplementary information - so that at least your results and interpretation are at least published.
Q5. You finish by remarking you have already published algorithms for 5-node subgraph generation and are embarking on testing your hypotheses (presumably both the utility of H_js and the existence of irreducible subgraphs) with these larger subgraphs. However no mention is made regarding the practicality of this approach beyond remarking that it is NP complete.

Below I discuss each section in turn, making recommendations as to their revision.

R1. Introduction
I recommend slightly restructuring the introduction to more precisely communicate 1) problem addressed 2) limitations of existing approaches and 3) a summary of your proposed approach and the prior work you drew on to develop it. In particular, it is important to communicate why network comparison is useful for biological network analysis, and how traditional graph alignment methodologies do not address this task.
I also note several specific issues:
i. line 50-52: Here you review a range of work relevant to undirected networks, and note that an 'integrated approach' is one of the well-developed methods to examine similarity of undirected networks. 'Integrated approach' means nothing to a ready unfamiliar with the 2011 paper, so you need to provide more information. For instance, if what was actually presented in that paper was a hybrid approach, then please say so! It also seems that these approaches are merely background - since your method is both alignment free and focused on directed networks. A more concise list of references without mentioning methods not relevant to the method would be appropriate here.

ii. Line 55-65: what exactly are 'graphlets' and 'directed graphlets' ? You say their use was introduced by Przulj and spend several sentences expanding on this, but a concise definition is not offered.

iii. In line 79-82. You state 'Many tools [20-24] have been developed to detect network motifs, but these tools may not be able to detect the complete set of motifs, because the predicted patterns are not statistically significant, i.e., p-values are larger than 0.05. This suggests that motif-finding tools have limitations, as they cannot enumerate all the network motifs embedded in a network.'. You then claim your method demonstrates this limitation [of other methods].

This point is quite critical and I recommend you revise this section to communicate it clearly. For example by: 1) explaining that many methods aim to detect 'statistically significant' motifs, and thus employ estimation heuristics such as random graphs, etc, and arbitrary thresholds such as p>0.05, 2) These methods often cannot detect all common motifs due to limitations of their approach 3) You propose an alternative approach based on existing work by yourselves and others. The fact that this work draws on ideas first presented by Mowshowitz is important, but the way you state this is weak (line 89-92) - and totally inaccessible for someone not already intimately familiar with the foundational literature of information-theoretic graph theory. It would be better to build up this hypothesis as part of a theoretical description in the methods section rather than try to skim it in the introduction.

The clarity of the introduction would also be improved by deferring any remarks regarding similarities and differences between your method and others to the discussion section, and refraining as much as possible from referring to the results section in your introduction. Instead, it is only necessary to summarise the rest of the paper - eg. 'In the next sections, we present our novel approach, its validation on simulated data, and its use to evaluate similarities amongst a range of biological and medically relevant networks'.


R2. Materials and Methods.

I was concerned to discover the methods section as currently written entirely lacked a clear *stand-alone* definition of the algorithm employed in this work. You must at the very least provide a concise protocol for how metrics designed for differentiating distributions (line 141-144) are applied to measure alignment-free graph similarity through a subgraph decomposition approach. Ideally you would do this by explicitly declaring how the subgraphs are enumerated in order to compute frequency distributions - a fairly straightforward task providing the section is slightly restructured.

Here, adopting a more traditional series of subsections such as definitions/algorithm/implementation/data for evaluation will help lead the reader from theoretical description to implementation and approach for evaluation.

i. line 114-128 describes download and preparation of real world data used for evaluation. Suggest you relocate to end of section, and introduce labels to denote each set of real world data to make it simpler to refer to it later in results.
ii. line 129-164 introduce definitions and outline the theoretical underpinnings. move to beginning of section
iii. line 165-174 describe the protocol for evaluation of the method on simulated data. This seems to make sense given the focus of the paper is this new method, but this protocol is not discussed until the second part of the results section. Here you can either reorder the results section to first discuss this validation step, or separate the results into two sections, each with their own methods/data prealbe: one focused on validation, and one focused on biological network analysis.

I also note the following specific issues and recommendations concerning the description of the algorithm:

R2.a. line 131. 'Each subgraph can be represented by a decimal' - Please clarify.. ie. is the decimal merely a label, or somehow derived from the adjacency matrix (ie a meaningful hash) ? If the former, then please say so.

R2.b. line 149-156 introduces the H_js metric by way of a description of H_kl - this should be made more accessible to those unfamiliar with these metrics by:
i. Formally introducing the H_js and H_kl quantitites as standing for Jensen-Shannon and Kullback-Leibler based metrics, possibly by also relocating the introduction of JS and KL (line 93-98) and expanding on how they can be applied to measure similarity between graphs.
ii. switching the order of sentences describing how you applied H_js (147-154) and its definition (154-162).
iii. Relocating your claim regarding the first use of H_js to measure network similarity through subgraph distributions to the discussion or conclusion.

R2.c. To address reviewer 3's comment, I recommend you ensure your revised algorithm description is sufficiently detailed to make clear how it differs from PatternFinder. Be assured I do understand your original intention of avoiding 'irrelevant content' by relegating PatternFinder's description to the appendix, but it is not appropriate to simply 'point out' in the methods section that PatternFinder is not designed for large-scale network analysis.. the reader should instead be already be aware from your introduction that any method that employs NP-complete enumeration will not provide a scalable solution for large-scale network analysis (as presumably required for biological networks), thus obviating the need of going into detail about patternfinder.

R2.d. In the results section you make several claims regarding 'irreducibility'. A brief formal definition of this should also be included in the methods section - presumably it refers to a subgraph not being composed of any composition of other subgraphs in a set (ie the basic unit of a fractal tiling, etc). A clear definition would help readers understand exactly what you mean later, eg. in line 244-247:
'For the 4-node subgraphs, eight subgraphs are not composed of any 4-node subgraphs : “id_14” (SIM), “id_28,” “id_74,” “id_76” (MIM), “id_280,” “id_328” (cascade), “id_392,” and “id_2184.” We considered these eight subgraphs to also exhibit the property of irreducibility.'
Taken out of context, this sentence seems circular (8 4-node subgraphs were not composed of any 4-node subgraphs), a clear definition of what is meant by this is necessary for it to be clear to the reader.

R3. Results.

As suggested earlier, the results section it best organised to clearly distinguish algorithm validation results and results of comparing biological network. One reviewer notes that limiting detailed discussion to one biological example could help improve clarity of results & discussion. I tend to agree.

a. You open this section stating 'we evaluated the performance of PatternFinder' .. - surely your mean to evaluate the performance of your patternfinder *based* algorithm for computing H_
b. Validation with simulated data. You note the algorithm performs perfectly in that it allows the computation of a metric capable of distinguishing a range of different classes of graph (as described in table 1).
i. Detailed description of igraph parameters is unnecessary in Table 1, particularly since you have already described the intended topology of each class. Please omit the final column and instead refer the reader to the igraph documentation.
ii. Table 2 combines both detailed presentatation of identified 3 and 4-node subgraphs, and their frequencies of observation.
* The legend should include the definition of all marks - ie '*' to indicate statistically significant according to the tool's own computation.
* there is probably no need to include each tool's own way of annotating the discovered subgraphs: instead simply report the frequencies, and refer the reader to the supplementary for these details. I would even suggest going further and including just a picture of the observed subgraph rather than its id number.

c. 3-node and 4-node subgraph biological network figures.
i. Please include detail in the legend explaining what the green coloured genes are, and why some genes are in red. I eventually inferred the reasons for these colourings only after noticing the genetic alterations box - that could also be listed in the legend rather than included in the diagram.
ii. In Figure 4 and 5 a grey box labelled 'scaffold' encloses the Dvl->b-catenin activation pathway. The legend and main body should explain why is this highlighted.
iii. It may be better to combine figures 2/3 and 4/5 into two figures, each with two sub-panels so that the respective network pairs can be visually compared.

d. Introduce subheadings for 'biological interpretation' - e.g. at line 288.
e. I do not consider the plots of normalized frequency distributions to be particularly informative on their own except to validate that H_js behaves as expected in detecting graphs with similar distributions of detected subgraphs. Consider combining all these plots to a single multi-panel figure, or showing exemplars to show how H_js relates to similar and different distributions.
f. heatmap visulisations
* Each column requires a distinct label if readers are to properly interpret these figures (see eg. Figure 11 HIF1-a and cAMP span three columns - what gene is represented by the intervening one ?)
* Visual scales should be shown indicating the mapping of shading to the range of values present in the heatmap


R4. Reproducibility
i. Code and scripts should be provided to perform the H_js calculation for networks in compliance with PeerJ requirements (https://peerj.com/about/author-instructions/#data-and-materials).
ii. ideally code/scripts used to create the simulated networks and compare them should also be provided. Whilst the arguments are effective for reporting it would have been better to simply provide the scripts used to generate results and figures.

Reviewer 1 ·

Basic reporting

The work describes an approach to determine network similarity in the contest of directed graphs. Overall the works is interesting and the description sufficiently clear.

The testing approach used by the author is acceptable although when exploring the function of genes embedded in certain motifs enrichment analysis should be used (including a measure of statistical validity) when making any claim on the functions of the genes. Perhaps, it would have been better to focus more on a single example (and only marginally mentions additional examples) rather than having a few rather unconnected examples that are only briefly described.

The approach used sounds conceptually very similar to motif analysis, and I'm a bit perplexed by the authors' effort to stress that it's very different. I see the analysis as complementary to classical motif analysis and I would encourage the author to include a more positive spin in the comparison rather than trying to stress the differences.

Experimental design

The testing approach sounds valid. I think some kind of flow diagram to describe the analysis would improve the readability but it's not necessary.

Validity of the findings

The algorithmic analysis sounds sound, but I'm a bit concerned about the validity of the biological findings. Picking one of the case studies and explore that more extensively would make the work more impactful.

Additional comments

The article is quite lengthy. A summary for the key findings would be definitely quite helpful.

Reviewer 2 ·

Basic reporting

The greatest flaw currently is the lack of appropriate coverage to current literature, in particular, the work of the Algo Dyn Lab, for example:

- Algorithmic complexity of motifs clusters superfamilies of networks (https://ieeexplore.ieee.org/document/6732768)
- Evaluating network inference methods in terms of their ability to preserve the topology and complexity of genetic networks (https://www.sciencedirect.com/science/article/abs/pii/S108495211630012X?via%3Dihub)
- Quantifying loss of information in network-based dimensionality reduction techniques (https://academic.oup.com/comnet/article-abstract/4/3/342/1745204)

Arix versions are available in most cases in case the authors find some of the above behind a firewall.

Experimental design

The paper is clear and methodologically sound, although some of the measures proposed are weak.

Validity of the findings

No comment

·

Basic reporting

Table 1 is very wordy. Please consider it to be Supplementary Note.
Please reshape Tables 4-6 to make them clearer.
Please increase the resolution of Figures 2-5 to make the content clearer.

Experimental design

It is interesting that the authors propose a method for analyzing and comparing molecular networks with an information entropy measure.

PatternFinder [32] the authors developed is also based on the network subgraph approach. However, the authors do not mention about PatternFinder in "Introduction". For example, at line 83, the authors write "we propose the network subgraph-based approach ..." but they do not refer to [32] in this line. Moreover, lines 86-92 "We hypothesize that network subgraphs are the fundamental building blocks ..." mostly overlap the lines in the "Network subgraphs (N-node subgraphs) vs. network motifs" section in [32]. The authors need to mention that (at least a part of) their approach already proposed in [32] in "Introduction".

The authors also need to clarify the difference between the method proposed in this paper and PatternFinder described in [32]. The only difference seems to be that they adopt H_js as the network similarity metric. Please clarify if other differences exist or not.

Validity of the findings

In Table 2, the authors compare the results between PatternFinder, acc-Motif, and Loto. But the comparison seems to be similar as Table 2 in [32] except that the input data are different. If the authors would like to point out the superiority of PatternFinder, they can do so only by referring to Table 2 in [32]. If they address other points, they need to clarify the difference between them.

It is difficult to understand Figure 1 and Table 3 because very similar topologies (sometimes the same topologies) of the networks are classified into different classes in Figure 1 but the classification accuracies are 100% in Table 3. For example, the "scale-free graph" in (a) is the same as the "Aging-Random Graph 2%". Please explain about Figure 1 and Table 3 in detail.

The authors state that Figures 2-3 and Figures 4-5 are embedded in 3-node and 4-node subgraphs, respectively. However, it is difficult to identify 3-node or 4-node subgraphs in the figures. It may be useful to show the locations of the subgraph "id"s of Table 4 in Figures 2-5.

Additional comments

no comment.

---

## Round 0.2 · accepted · Accept

Thank you for addressing the earlier questions and concerns.

·

Basic reporting

The paper has been revised according to reviewers' comments. The reviewer has no further comments on the revised version of the paper.

Experimental design

No further comments.

Validity of the findings

No further comments.